# DEMONet: A Dual-channel Multi-omics Integration Hypergraph Network for Cancer Gene Identification

## Abstract

Identifying cancer driver genes is a critical challenge in cancer genomics. While hypergraph neural networks are powerful tools for identifying cancer genes by modeling higher-order functional relationships, they face a critical limitation in integrating multi-omics features. To address this, we propose DEMONet, a Dual-channel Multi-Omics Integration Hypergraph Network. DEMONet enhances multi-omics integration through three synergistic modules: (1) a tree-based sparse encoder that transforms raw multi-omics features into a robust, structured representation; (2) a biologically-informed node-weighted hypergraph convolutional layer to capture gene importances within functional hyperedges; and (3) a dual-channel architecture to prevent information interference between different hypergraph sources before final fusion. Benchmark results demonstrate that DEMONet outperforms existing state-of-the-art methods, improving AUROC by 1.9% and AUPRC by 2.3% over prior methods. Its generalization and robustness are further validated on two independent test sets. Analysis of multiple independent functional genomics data validated the significant biological associations between the DEMONet-predicted top candidate genes and cancer genes. Furthermore, TCGA survival analysis further reveals that 16 novel cancer genes identified by DEMONet are significantly associated with patient outcomes, highlighting the potential of our model to discover actionable targets for cancer research.

## 1 Introduction

Cancer arises from the progressive accumulation of somatic genomic alterations that deregulate cellular processes and endow cells with malignant capabilities (Hanahan & Weinberg, 2011). Identifying the driver genes responsible for these changes is a central goal in cancer genomics, as a comprehensive catalog of cancer driver genes is essential for precision diagnostics and targeted therapies (Garraway & Lander, 2013; Alexandrov et al., 2013). Although large-scale sequencing projects have produced invaluable resources like the COSMIC Cancer Gene Census (CGC) and the Network of Cancer Genes (NCG) (Sondka et al., 2018; Repana et al., 2019), these catalogs remain far from complete. The sheer diversity and complexity of the cancer genome mean that many driver genes, particularly those mutated at lower frequencies or acting in a context-dependent manner, are yet to be discovered (Lawrence et al., 2014). Thus, the accurate identification of cancer genes remains a critical challenge.

In response, the computational biology community has developed a sophisticated suite of tools for cancer gene prioritization, which have evolved through several distinct methodological paradigms. The earliest approaches were mutation-centric, with methods like MutSigCV (Lawrence et al., 2013) employing statistical models to identify genes mutated more frequently than expected by chance, and later methods like 20/20+ (Tokheim et al., 2016) and DORGE (Lyu et al., 2020) engineering these statistics into machine learning features. A subsequent paradigm shift leveraged biological networks, such as the graph neural network-based EMOGI (Schulte-Sasse et al., 2021) and MTGCN (Peng et al., 2022). Most recently, the field has begun to embrace higher-order functional modules, recognizing that genes often act in coordinated groups. For example, DISHyper (Deng et al., 2024) utilizes hypergraph neural networks to learn powerful gene representations from the higher-order functional association of annotated gene sets, such as those from pathway and ontology databases.

Despite their advancements, their methods still face significant limitations. First, multi-omics features exhibit a high degree of heterogeneity, manifested in a wide range of skewness and kurtosis values and large standard deviations (details in Appendix A and Appendix Figure 3). This characteristic poses significant challenges for their effective integration with network topologies. This often forces models to choose between rich topological context and powerful but incompatible node features, and our preliminary experiments confirm that naive fusion can degrade performance. Second, current hypergraph models typically assume node homogeneity within a hyperedge, treating all genes in a pathway as equally important. This simplification overlooks the context-dependent and hierarchical nature of biological systems. For example, a gene like MYC can act as a key regulator of cell proliferation, yet it also functions as a downstream target of mitogenic signaling pathways in other contexts. Existing models fail to capture the gene's contextual functional specificity, thereby diluting the learned signals. Finally, the strategy for fusing disparate hypergraph sources via simple concatenation is suboptimal, as it may lead to representational interference, where unique topological patterns from different sources become obscured by one another.

To overcome these challenges, we propose DEMONet, a novel hypergraph framework that integrates three synergistic modules. First, to effectively integrate heterogeneous multi-omics features, DEMONet introduces a tree-based multi-omics sparse encoding module. This module transforms the irregular heterogeneous features into a sparse, high-dimensional representation that is more amenable to deep learning models. Second, to model the heterogeneous importance of genes within a functional hyperedge, we develop a biologically-informed node-weighted hypergraph convolutional layer. Third, to mitigate representational interference, DEMONet employs a dual-channel architecture. This design learns representations from two hypergraphs in parallel and then fuses their final embeddings for prediction.

Our contributions are highlighted as follows:

- We propose a novel hypergraph framework, DEMONet, that systematically addresses key limitations in cancer gene identification.
- We demonstrate through extensive experiments that DEMONet significantly outperforms state-of-the-art methods on both pan-cancer and two independent test sets.
- We systematically validate the effectiveness and synergy of our three core modules through comprehensive ablation studies.
- We validate the biological and clinical relevance of DEMONet-predicted cancer genes using independent functional genomics and The Cancer Genome Atlas (TCGA) patient survival data. This real-world validation underscores the practical utility of our approach in cancer research.

## 2 RELATED WORKS

The identification of cancer genes can be conceptualized as a classification problem. The objective is to accurately distinguish the relatively small set of true cancer-driving genes from the vast background of passenger genes. Research in this area encompasses several major methodological paradigms.

Early approaches are primarily feature-based, relying on statistical properties derived directly from sequencing data. A foundational method in this category is MutSigCV (Lawrence et al., 2013). Building upon this, subsequent methods incorporate machine learning. For instance, 20/20+ and DORGE engineer a wide array of features, including mutation patterns and epigenetic markers, to train classifiers such as elastic net–based logistic regression, thereby transitioning from rule-based identification to predictive modeling (Tokheim et al., 2016; Lyu et al., 2020; Lee et al., 2024).

A second major paradigm involves leveraging the context of biological networks. These methods are grounded in the "guilt-by-association" principle, positing that genes involved in the same disease are often functionally linked. For example, HotNet2 (Leiserson et al., 2015) and Hierarchical HotNet (Reyna et al., 2018) use heat diffusion models on a protein-protein interaction (PPI) network to identify subnetworks significantly enriched with cancer mutation signatures.

With the development of deep learning, Graph Neural Networks (GNNs) represent a dominant force in this field, offering a more powerful way to learn representations from complex network struc-

tures. EMOGI (Schulte-Sasse et al., 2021) employs a GNN to integrate multi-omics features with a PPI network. MTGCN (Peng et al., 2022) enhances this by introducing a PPI network reconstruction task, using multi-task learning to enrich the learned gene representations. Others employ more advanced architectures, such as graph attention networks (MODIG (Zhao et al., 2022)) or multi-network contrastive learning (MNGCL (Peng et al., 2024)), to fuse information from diverse biological sources. Distinct from these pairwise network models, a new paradigm emerges that focuses on higher-order functional associations. These methods recognize that genes often function in complex, multi-member groups rather than simple pairs. For example, DISHyper (Deng et al., 2024) utilizes a hypergraph neural network to model these higher-order relationships in annotated gene sets explicitly. However, critical limitations exist in current hypergraph-based approaches that we aim to resolve.

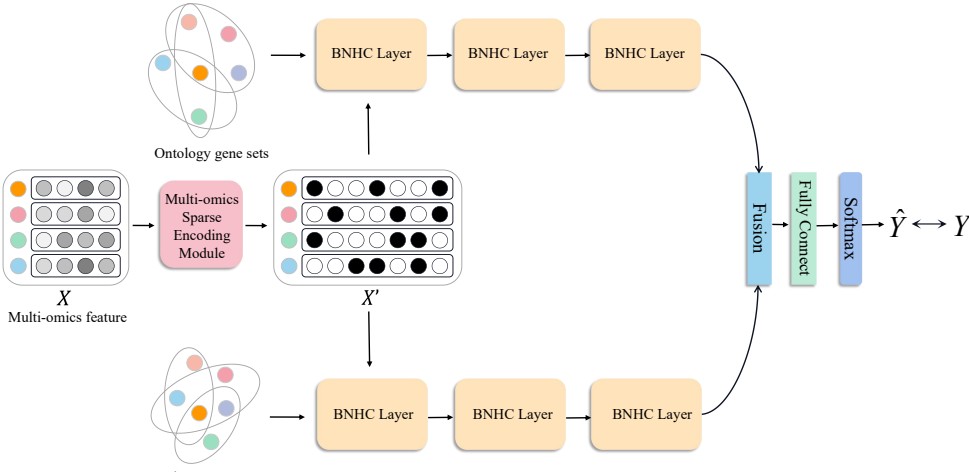

Figure 1: Overview of the DEMONet framework. DEMONet predicts cancer driver scores by processing two hypergraphs within a dual-channel architecture. Input multi-omics features are first unified via a tree-based sparse encoding module. Each channel then employs a biologically-informed node-weighted hypergraph convolutional layer to learn gene representations. The resulting embeddings are fused for the final prediction.

## 3 METHODOLOGY

The architecture of the DEMONet is designed to overcome the primary obstacles in computational cancer genomics. It comprises three core components: (1) a multi-omics sparse encoding (MSE) module, which transforms irregular cancer multi-omics features into a feature space compatible with deep learning through a tree model; (2) a biologically-informed node-weighted hypergraph convolutional (BNHC) layer, which introduces prior node weights to guide the message-passing process; and (3) a dual-channel hypergraph network architecture that preserves the informational integrity of disparate hypergraph sources before final fusion. The overall workflow is depicted in Figure 1.

### 3.1 MULTI-OMICS SPARSE ENCODING MODULE

For any given gene, its multi-omics features (e.g., somatic mutation frequency, DNA methylation) form a heterogeneous vector with diverse statistical properties, exhibiting highly non-Gaussian distributions (Appendix A and Appendix Figure 3). While neural networks do not make explicit distributional assumptions, their training dynamics and performance are highly sensitive to the scale and shape of input features. Therefore, directly inputting these raw, irregular features into a deep hypergraph neural network can impede the training process, leading to models that struggle to effectively handle disparate feature scales and are less efficient at learning complex nonlinear interactions. To address this problem, we propose the MSE module. The objective of this module is to perform a non-linear transformation that maps the original multi-omics feature space into a high-dimensional, sparse, and structured latent space.

Our methodology leverages a Random Forest to achieve this feature transformation. A forest consisting of $\mathcal{T}$ decision trees is first trained using only the training set data. Once trained, we utilize its learned structure as a feature encoder. For any given gene g with its raw feature vector $\mathbf{X}_g$, we pass it through every tree in the forest to obtain a set of terminal leaf node indices. The final sparse feature representation, $\mathbf{X}_g^{'}$, is generated by concatenating the one-hot encodings of these leaf indices. This process can be formally expressed as:

$$\mathbf{X}_g^{'} = \bigoplus_{t=1}^{T} \text{one-hot}(l_t(\mathbf{X}_g)) \tag{1}$$

where $l_t(\mathbf{X}_g)$ is the index of the leaf node that the feature vector $\mathbf{X}_g$ lands in for tree $\mathcal{T}_t$, and $\bigoplus$ denotes the concatenation operation. The resulting vector $\mathbf{X}_g^{'}$ is a high-dimensional and sparse binary representation. This new representation implicitly encodes higher-order feature interactions by converting complex numerical relationships into a robust binary feature format, making it more amenable to deep learning architectures.

## 3.2 BIOLOGICALLY-INFORMED NODE-WEIGHTED HYPERGRAPH CONVOLUTIONAL LAYER

Annotated gene sets, such as those from ontology gene sets or pathway gene sets, naturally capture the higher-order functional relationships among multiple genes. To leverage this gene functional association information, we model it as a hypergraph, denoted as $\mathcal{G} = (\mathcal{V}, \mathcal{E}, \mathbf{W}_e)$. The set of nodes $\mathcal{V} = \{v_1, \ldots, v_n\}$ represents the n genes, and the set of hyperedges $\mathcal{E} = \{e_1, \ldots, e_m\}$ represents the m annotated gene sets. The structure of the hypergraph is mathematically described by its incidence matrix $\mathbf{H} \in \mathbb{R}^{n \times m}$, where an entry $\mathbf{H}(v, e) = 1$ if gene v is a member of gene set e, and 0 otherwise. Furthermore, $\mathbf{W}_e$ is a diagonal matrix containing the weights assigned to each hyperedge.

A critical limitation of conventional hypergraph neural networks is their assumption of uniform node importance within a hyperedge. This implies that all genes in a biological pathway contribute equally. However, this is a stark oversimplification of biological reality, as the roles of genes are highly context-dependent. For instance, a transcription factor might play a pivotal regulatory role, while other genes in the same pathway act as downstream effectors. To overcome this, we introduce the BNHC layer. The BNHC layer assigns biologically-informed prior weights to nodes within each hyperedge, thereby guiding the message aggregation process to focus on functionally central genes.

**Biologically-informed node weight calculation.** The core of the BNHC layer is the computation of a unique prior weight for each node $v$ within a specific hyperedge $e$. Due to the distinct intrinsic properties of our hypergraph sources, this biologically-informed weighting strategy is specifically applied to the ontology gene sets, while pathway gene sets receive uniform weights (details in Appendix B). This weight, denoted as $w_{ve}$, quantifies the node's importance in the context of that biological function. We derive this weight by assessing whether the local connectivity of gene $v$ with other members of $e$ in HumanNet v3 (Kim et al., 2022) is significantly higher than expected by chance. If the local connectivity is higher than expected, it means that the gene is likely to play a key role in the biological process. This statistical enrichment, modeled via the hypergeometric distribution, serves as a powerful proxy for gene importance (Fang et al., 2012). Specifically, we treat the genes in HumanNet v3 as the population, the genes in hyperedge $e$ as the sample, and the neighbors of $v$ in the HumanNet v3 as the items of interest. We then calculate the expectation $E[k_v^e]$ under this null hypothesis. The raw enrichment score is then transformed into a stable weight:

$$w_{ve} = \log_2 \left( (k_v^e - E[k_v^e]) \cdot \mathbb{I}(k_v^e > E[k_v^e]) + 2 \right) \tag{2}$$

where $k_v^e$ is the number of connections observed in HumanNet v3 between gene $v$ and other genes in hyperedge $e$, and $\mathbb{I}$ is the indicator function. The constant 2 ensures a baseline weight of 1 (for non-enriched genes, where the difference is zero or negative) and produces weights significantly greater than 1 for genes with higher-than-expected connectivity.

**Node-weighted aggregation.** The BNHC layer then performs the first stage of message passing: node-to-hyperedge aggregation. This process is guided by our pre-computed, biologically-informed node weights $w_{ve}$, which act as local importance scores. The representation of hyperedge $e$ at layer $(l)$, denoted as $\mathbf{h}_e^{(l)}$, is constructed by a weighted aggregation of its constituent node features:

$$\mathbf{h}_e^{(l)} = \sum_{v \in e} w_{ve} \cdot \mathbf{W}_{n \to e}^{(l)} \mathbf{h}_v^{(l-1)} \tag{3}$$

where $\mathbf{h}_v^{(l-1)}$ is the feature vector of the node $v$ from the previous layer (with $\mathbf{h}_v^{(0)} = \mathbf{X}'$), and $\mathbf{W}_{n \to e}^{(l)}$ is a learnable linear transformation for the $(l)$-th layer. This weighted aggregation ensures that the resulting hyperedge representation is dominated by its most functionally relevant members, leading to more meaningful and biologically plausible embeddings.

**Supervised Hyperedge-to-Node Aggregation.** The second stage of the BNHC layer is hyperedge-to-node aggregation, where each node updates its representation by integrating information from its incident hyperedges.

To enhance the learning process in a high-noise biological context, we adopt a supervised hyperedge weighting scheme to prioritize hyperedges with higher known relevance to cancer. This strategy, whose utility has been demonstrated in prior hypergraph-based methods (Deng et al., 2024), acts as a strong inductive bias. Specifically, using the labels of training set exclusively, we compute a global weight $\mathbf{w}_e$ for each hyperedge $e$ as the proportion of known cancer driver genes it contains. This weight is designed to reflect the cancer-specific relevance of the hyperedge. The updated node representation $\mathbf{h}_v^{(l)}$ is computed as:

$$\mathbf{h}_v^{(l)} = \sigma \left( \sum_{e \in E_v} w_e \cdot \mathbf{W}_{e \to n}^{(l)} \mathbf{h}_e^{(l)} + \mathbf{h}_v^{(l-1)} \right) \tag{4}$$

$$w(e) = \frac{\sum_{v \in V} \mathbf{H}(v,e) f(v, V_d)}{\sum_{v \in V} \mathbf{H}(v,e)} \tag{5}$$

where $E_v$ is the set of hyperedges containing node $v$, $\mathbf{h}_e^{(l)}$ is the hyperedge representation from the first stage, and $\mathbf{W}_{e \to n}^{(l)}$ is another learnable weight matrix. The $\sigma$ is a non-linear activation function. The $V_d$ denotes the known cancer gene in the training set, and $f(v, V_d)$ is used to indicate whether gene $v$ belongs to $V_d$. This supervised weighting approach may have potential drawbacks, but our model exhibits strong generalization ability on two independent test sets and demonstrates its ability to identify cancer genes in validation analyses of prediction results. This dual-stage, weighted message-passing mechanism allows the model to learn a more expressive and biologically plausible representation of genes.

### 3.3 DUAL-CHANNEL HYPERGRAPH NETWORK ARCHITECTURE

To effectively leverage hypergraphs from disparate sources, we introduce a dual-channel hypergraph network architecture. The rationale behind this design is that different databases, such as ontology gene sets and pathway gene sets, capture distinct facets of gene function with unique topological properties. The naive approach of merging them into a single overall hypergraph may result in one type of information obscuring or dominating the other. Our dual-channel architecture mitigates this by processing hypergraphs from different sources in parallel, allowing for specialized learning before final fusion.

Specifically, for the two hypergraph sources—ontology gene sets and pathway gene sets—our framework instantiates two independent hypergraph network channels, each composed of a stack of the three BNHC layers described previously. One channel is dedicated to processing the hypergraph constructed from ontology gene sets, $\mathcal{G}_{Ontology}$, while the other processes the hypergraph derived from pathway gene sets, $\mathcal{G}_{Pathway}$. The initial node features, $\mathbf{X}'$, from the MSE module are shared and serve as input to both channels. Each channel then learns a specialized node embedding matrix that captures the functional context specific to its data source. The ontology channel produces an ontology-centric embedding, $\mathbf{Z}_{Ontology} = \text{Channel}_{Ontology}(\mathbf{X}', \mathcal{G}_{Ontology})$, while the pathway channel generates a pathway-centric embedding, $\mathbf{Z}_{Pathway} = \text{Channel}_{Pathway}(\mathbf{X}', \mathcal{G}_{Pathway})$.

After the final layer of each channel, the two specialized embedding matrices are fused to generate a comprehensive gene representation. For this fusion, we employ a concatenation operation, which preserves the complete information learned from both contexts:

$$\mathbf{Z}_{final} = \mathbf{Z}_{Ontology} \bigoplus \mathbf{Z}_{Pathway} \tag{6}$$

$$\hat{Y} = Softmax(\mathbf{Z}_{final} \mathbf{W}_{final} + \mathbf{b}) \tag{7}$$

where $\bigoplus$ denotes concatenation, $\mathbf{W}_{final}$ is a learnable weight matrix and $\mathbf{b}$ is the bias vector. Finally, the integrated representation $\mathbf{Z}_{final}$ is input into a fully connected layer and a softmax function to obtain the cancer gene probability score for each gene. This architecture ensures that the model can learn from different biological function contexts without being disturbed, thereby obtaining a more robust and accurate final prediction.

**(A)**                                                                 **(B)**

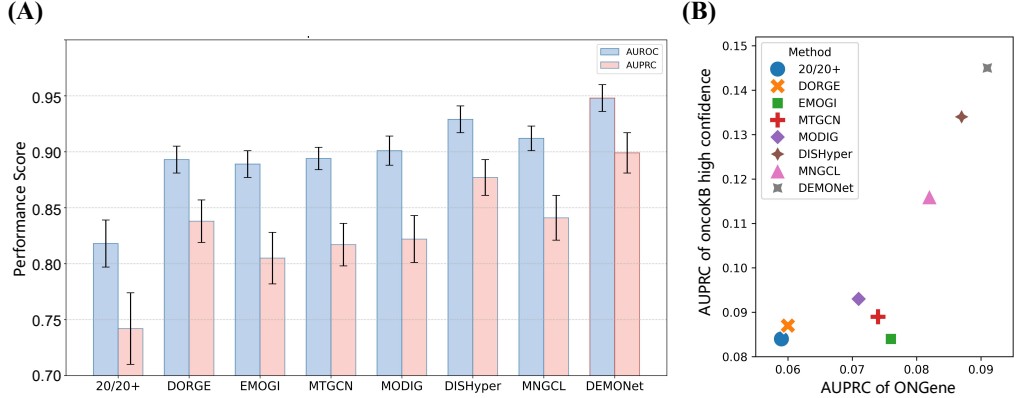

Figure 2: Benchmarking results of DEMONet. **(A)** Performance comparison of DEMONet and seven cancer gene identification methods. Bars represent the mean performance, and error bars indicate the standard deviation across five independent runs. **(B)** Performance comparison of DEMONet and other methods on two independent test sets.

## 4 EXPERIMENT

### 4.1 DATA SOURCE AND BASELINES

**Data source.** The input for DEMONet primarily consists of two components: multi-omics features for each gene and two collections of annotated gene sets. Furthermore, we use a curated set of known cancer and non-cancer genes as labeled data to train and evaluate our model.

The multi-omics features for each gene are compiled from multiple sources to provide a comprehensive biological profile. This includes somatic mutation, DNA methylation, and gene expression data derived from approximately 8,000 tumor samples across 16 cancer types from TCGA study (Schulte-Sasse et al., 2021). We augment this with 10 system-level properties, such as gene duplication status and miRNA interactions, as described by Nulsen et al (Nulsen et al., 2021). Additionally, to capture network context, we include 16-dimensional topological features for each gene, which are generated by applying the DeepWalk (Perozzi et al., 2014) algorithm to a PPI network. The concatenation of these features results in a final 74-dimensional multi-omics feature vector for each gene.

The annotated gene sets, which form the basis of our hypergraphs, are sourced from the Molecular Signatures Database (MSigDB) (Liberzon et al., 2015). We utilize two collections: (1) ontology gene sets, comprising 14,905 sets from the Gene Ontology (GO) (Consortium, 2004) and Human Phenotype Ontology (HPO) databases (Köhler et al., 2021), and (2) pathway gene sets, consisting of 5,721 pathways curated from expert databases such as Reactome (Fabregat et al., 2018) and KEGG (Kanehisa & Goto, 2000).

For model training and evaluation, we compile a high-confidence set of labeled genes. Positive samples (known cancer driver genes) are collected from three authoritative databases: NCG (v6.0) (Repana et al., 2019), COSMIC Cancer Gene Census (v91) (Sondka et al., 2018), and DigSEE (Kim et al., 2013). Negative samples (non-cancer genes) are stringently defined by iteratively removing any genes present in NCG, COSMIC CGC, DigSEE, and the KEGG cancer pathway gene sets. This process yields a final dataset of 796 positive and 2,187 negative gene samples.

**Baseline and implementation.** We evaluate the performance of DEMONet against a comprehensive set of seven state-of-the-art baseline methods, which span the major methodological paradigms

in cancer gene identification. These include 20/20+ (Tokheim et al., 2016), DORGE (Lyu et al., 2020), EMOGI (Schulte-Sasse et al., 2021), MTGCN (Peng et al., 2022), MODIG (Zhao et al., 2022), DISHyper (Deng et al., 2024) and MNGCL (Peng et al., 2024). Further details regarding the implementation of DEMONet and the experimental settings are provided in Appendix C. Our source code and all data used in this study are made publicly available in the Supplementary Materials to ensure reproducibility.

## 4.2 PERFORMANCE COMPARISON WITH STATE-OF-THE-ART METHODS

To evaluate the performance of DEMONet, we conduct a comprehensive benchmark against seven state-of-the-art methods for cancer gene identification. We assess all models based on the area under the receiver operating characteristic curve (AUROC) and the area under the precision-recall curve (AUPRC).

Our framework shows substantial performance gains over methods from all major paradigms (Figure 2A). When compared to traditional machine learning models that rely on manually engineered features, such as 20/20+ and DORGE, DEMONet achieves a significant improvement of at least 5.5% in AUROC and 6.1% in AUPRC ($P$-value $< 0.05$ by one-sided Wilcoxon signed-rank test). Similarly, DEMONet outperforms representative biological network-based methods. Against EMOGI, MT-GCN, and MODIG, our model yields performance boosts of no less than 4.6% in AUROC and 6.6% in AUPRC ($P$-value $< 0.05$). Notably, even when benchmarked against MNGCL, a method that integrates multiple biological networks, DEMONet exhibits a remarkable advantage, with relative increases of 3.6% in AUROC and 5.8% in AUPRC ($P$-value $< 0.05$). Most importantly, compared to DISHyper, the most comparable hypergraph-based baseline, DEMONet still delivers significant enhancements of 1.9% in AUROC and 2.3% in AUPRC ($P$-value $< 0.05$). Collectively, these results establish the superior performance of DEMONet.

To further assess the generalization capability of DEMONet and its ability to identify novel cancer genes, we evaluate its performance on two independent test sets of cancer genes curated from the OncoKB (Chakravarty et al., 2017) and ONGene (Liu et al., 2017) databases. These sets, comprising 313 and 382 cancer genes respectively, are not used during model training, thus providing an unbiased testbed. For this evaluation, we use AUPRC as the primary metric, treating the genes in each independent set as true positives and all other genes as negatives. As illustrated in Figure 2B, DEMONet consistently achieves the best performance on both independent datasets, underscoring its strong generalization power. Although the absolute AUPRC values are expectedly low due to the high imbalance between positive and negative samples in this challenging task, the relative performance gains are substantial. Compared to all other methods, DEMONet demonstrates a relative improvement of at least 6% and 4.6% on the OncoKB and ONGene test sets, respectively. This superior performance on unseen data strongly suggests that DEMONet is not merely memorizing patterns in the training set but has learned a more fundamental and generalizable representation of cancer gene properties, making it a more reliable tool for discovering novel cancer genes.

Table 1: Effectiveness of the MSE Module. The best performance is highlighted in **bold**.

| Model Variant | Feature Type | Ontology | | Pathway | |
| --- | --- | --- | --- | --- | --- |
| | | AUROC | AUPRC | AUROC | AUPRC |
| (1) DEMONet w/o MSE | One-hot (No features) | 0.911 | 0.843 | 0.918 | 0.832 |
| (2) RF + Raw Features | Raw Multi-omics | 0.922 | 0.828 | 0.922 | 0.828 |
| (3) DEMONet-Raw | Raw Multi-omics | 0.841 | 0.735 | 0.881 | 0.774 |
| (4) MLP + Sparse Features | Sparse Encoded | 0.916 | 0.833 | 0.916 | 0.833 |
| (5) DEMONet | Sparse Encoded | **0.939** | **0.885** | **0.940** | **0.886** |

## 4.3 ABLATION STUDIES

To rigorously evaluate the contribution of each module within the DEMONet framework, we conduct a series of comprehensive ablation studies. These experiments are designed to systematically isolate and evaluate the impact of the multi-omics sparse encoding module, the node-weighted hypergraph convolution layer, and the dual-channel architecture.

### 4.3.1 Effectiveness of the Multi-Omics Sparse Encoding Module

First, we investigate the impact of our proposed MSE module. We evaluate its effectiveness on both the ontology and pathway gene sets by comparing the full DEMONet model against four baseline variants: (1) DEMONet w/o MSE, which uses only one-hot vectors as input features; (2) RF + Raw Features, a standard Random Forest classifier trained on the raw multi-omics features; (3) DEMONet-Raw, where the raw multi-omics features are directly fed into our model; and (4) MLP + Sparse Features, a Multi-Layer Perceptron trained on our sparsely encoded features.

The results, presented in Table 1, reveal several key insights. A direct comparison between DE-MONet (5) and DEMONet-Raw (3) shows that naively feeding raw features into the model degrades performance, which we attribute to the irregular distribution and high bias of the multi-omics data being incompatible with the deep learning architecture. Furthermore, comparing DEMONet (5) with the MLP-based approach (4) and the RF classifier (2) demonstrates that neither the sparse encoding nor the raw features alone are sufficient for optimal performance when used with simpler classifiers. It is the powerful synergy between the MSE module's structured feature representation and the DEMONet's ability to learn from higher-order topology that yields the best results. These findings underscore the critical importance of the MSE module in bridging the gap between complex biological data and deep learning models.

Table 2: Ablation study of BNHC.

| Components | | Performance | |
| --- | --- | --- | --- |
| MSE | BNHC | AUROC | AUPRC |
| 0 | 0 | 0.896 | 0.827 |
| 0 | 1 | 0.911 | 0.843 |
| 1 | 0 | 0.931 | 0.876 |
| 1 | 1 | **0.939** | **0.885** |

Table 3: Ablation study of the dual-channel architecture.

| Model / Data Configuration | AUROC | AUPRC |
| --- | --- | --- |
| DEMONet - Ontology Channel Only | 0.939 | 0.885 |
| DEMONet - Pathway Channel Only | 0.940 | 0.886 |
| DEMONet - Concatenated Hypergraph | 0.939 | 0.886 |
| DEMONet - Dual-Channel | **0.948** | **0.899** |

### 4.3.2 Impact of the Biologically-Informed Node-Weighted Hypergraph Convolutional Layer

Next, we evaluate the contribution of the BNHC layer to the ontology gene set, since our prior weight calculation scheme is designed for the ontology gene set. We conduct this analysis under two conditions: with and without the MSE module, to test the robustness of the BNHC layer's contribution.

As shown in Table 2, the inclusion of the BNHC layer consistently improves model performance. In the absence of the MSE module, adding the BNHC layer boosts AUROC and AUPRC by 1.5% and 1.6%, respectively. Even when integrated into the full DEMONet model, where rich node features already provide a strong predictive signal, the BNHC layer still provides performance gains of 0.8% in AUROC and 0.9% in AUPRC. This consistent improvement demonstrates that explicitly modeling the heterogeneous importance of genes within functional pathways is a key factor in enhancing the model's predictive accuracy.

### 4.3.3 Analysis of the Dual-Channel Architecture

Finally, we validate the effectiveness of our dual-channel architecture. We compare the full DE-MONet model against three alternatives: (1) a model using only the ontology gene set channel, (2) a model only using the pathway gene set channel, and (3) a single-channel model trained on a naively concatenated hypergraph of both gene sets.

The results in Table 3 clearly illustrate the benefits of the dual-channel design. While both individual channels show strong predictive power, simply concatenating the two hypergraph sources does not lead to any performance improvement. This suggests that naive fusion can lead to the distinct topological patterns of each data source being diluted. In contrast, our dual-channel architecture, which learns specialized representations before fusion, achieves a performance increase, improving AUROC by 0.8% and AUPRC by 1.3% over the best single-source model. This result validates our hypothesis that processing disparate biological knowledge sources in parallel is a more effective strategy for information integration.

## 4.4 Validation and Functional Characterization of Predicted Cancer Genes

To validate the ability of DEMONet to prioritize biologically relevant cancer genes, we first assess our predictions against a high-confidence set of genes from the CancerMine database (Lever et al., 2019). Our analysis reveals that top-ranked genes from DEMONet are highly enriched with these literature-supported cancer genes. Specifically, 86 of the top 200 prioritized genes are present in this high-confidence set (Appendix Figure 4). Furthermore, the decile analysis shows that the high-confidence set is significantly enriched in the first decile, exhibiting a clear trend of decreasing frequency in subsequent deciles (Appendix Figure 4). More detailed analysis results are in the Appendix D. These results suggest that the genes ranked highest by our model are substantially more likely to be genuine cancer-related genes.

To further investigate the biological significance of our predictions, we analyze the DEMONet-predicted top-ranked 100 cancer genes (PCGs) for enrichment in three independent functional genomics datasets representing key cancer-driving mechanisms: transposon-based gene inactivation, oncogenic gene fusions, and epigenetic regulation. We compare the enrichment of our PCGs against that of known cancer genes (KCGs) and neutral genes (NGs). We find that our PCGs, much like KCGs, are significantly enriched in all three functional categories, whereas neutral genes show no such association (Appendix Figure 4). This demonstrates that our predicted genes share hallmark properties with established cancer drivers operating through diverse molecular mechanisms. Detailed statistical results for each analysis are available in Appendix E.

## 4.5 Clinical Relevance of Novel Candidate Cancer Genes Revealed by Survival Analysis

To demonstrate the practical value of DEMONet, we analyze the top candidate genes prioritized by DEMONet. We identify 16 potential novel cancer genes (novelCGs, Appendix Table 4) within our top 100 predictions that are absent from major cancer gene databases (CancerMine, NCG, COSMIC CGC). Subsequently, to assess the association between the expression of these novelCGs and patient prognosis, we perform a comprehensive survival analysis using the GEPIA2 platform (Tang et al., 2019). The analysis is conducted across 33 cancer types from TCGA.

The results provide strong evidence for the clinical significance of our predictions. As shown in Appendix Figure 5, the expression of all 16 novelCGs is significantly associated with patient prognosis in at least one cancer type. Notably, 15 of these genes, including *ITGB3*, *NRP1*, and *COL1A2*, show significant prognostic power in two or more cancer types, suggesting their broad involvement in cancer progression. In addition, *ITGB3* has also been mechanistically demonstrated to play an important role in the metastasis of breast cancer (Fuentes et al., 2020). Down-regulation of *NRP1* promotes breast cancer progression and may be a target for new strategies for the treatment of breast and other cancers (Dong et al., 2021). *COL1A2* has also been found to have an inhibitory effect on colorectal cancer cell proliferation, migration, and invasion(Yu et al., 2018). The above results demonstrate the potential of DEMONet to identify clinically actionable targets for future cancer research.

## 5 Conclusion

In this paper, we introduce DEMONet, a novel dual-channel hypergraph network designed to address key challenges in cancer gene identification. DEMONet advances the state-of-the-art by synergistically integrating three key innovations: a multi-omics sparse encoder to robustly process complex biological features, a biologically-informed node-weighted convolution to capture gene heterogeneity within a hyperedge, and a dual-channel architecture to effectively fuse disparate hypergraphs. Extensive experiments demonstrate that DEMONet significantly outperforms a wide range of existing methods and exhibits superior generalization on two independent test sets. Our comprehensive ablation studies rigorously validate the critical contribution of each proposed component. Most importantly, we show that the novel candidate genes prioritized by DEMONet are not only highly enriched in diverse cancer-driving mechanisms but also significantly associated with patient survival outcomes. These findings underscore the practical utility of DEMONet as a powerful tool for discovering clinically relevant and actionable targets for future cancer research.

ETHICS STATEMENT

All authors have read and adhered to the ICLR Code of Ethics.

REPRODUCIBILITY STATEMENT

The source code for the DEMONet model is provided as supplementary material. Our implementation is primarily built upon standard libraries, including PyTorch and PyTorch Geometric.

Details regarding the model's architecture, such as the Random Forest configuration in the MSE module, the hidden dimensions of the BNHC layers, the specific hyperparameters used for training DEMONet, and the evaluation protocol for all baseline models, are described in Appendix C.

All datasets used in this study are publicly available. A comprehensive description of data sources, pre-processing steps, and the construction of our training and validation sets is provided in Section 4.1.

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

## A    STATISTICAL ANALYSIS OF MULTI-OMICS FEATURE DISTRIBUTIONS

We conduct a statistical analysis to characterize the distributions of five distinct sets of multi-omics features: Somatic Mutation, DNA Methylation, Gene Expression, System-level properties, and Topological features. By calculating the skewness and kurtosis for each gene's feature vector, we aim to quantify the shape and tail behavior of multi-omics features. To visually illustrate the statistical heterogeneity of the multi-omics features, we standardize all features via z-score transformation and plot their distributions against a standard normal distribution (Figure 3). The analysis reveals significant statistical heterogeneity across all feature sets, with substantial deviations from a normal distribution, which motivates the need for models robust to such data characteristics.

The mutation features exhibit the most extreme statistical properties, consistent with their sparse, count-based nature. They are characterized by an exceptionally high average skewness of 19.04 and a staggering mean kurtosis of 1175.68. Crucially, the standard deviations of these metrics (21.78 for skewness and 2417.19 for kurtosis) are even larger than their respective means, with the maximum observed kurtosis reaching nearly 10,000. This indicates that the mutation data is sparse and dominated by features with extreme outliers.

In comparison, gene methylation and gene expression features, while also markedly non-Gaussian, exhibit more contained distributions. Both sets are strongly right-skewed (mean skewness of 2.64 and 2.78, respectively) and are highly leptokurtic, indicating a prevalence of heavy tails in these molecular measurements. They differ, however, in their internal consistency. Methylation features show a greater diversity in their tail behavior, with a kurtosis standard deviation (9.41) that is larger than its mean (9.14). Gene expression features, conversely, are more uniformly heavy-tailed, with a lower relative standard deviation in kurtosis (mean 12.09, std 5.47).

The final two feature sets capture higher-level properties of the genes. The system-level properties are a mixed feature, containing both left-skewed (min skewness -1.09) and strongly right-skewed features (max skewness 46.47), leading to a large standard deviation of skewness (14.22). The topological features, which represent the structural properties of each gene within biological networks, are on average left-skewed (mean -1.43). However, this average conceals a balanced mixture of left- and right-skewed distributions, as evidenced by a wide range from -9.37 to 9.93. Both of these derived feature sets also exhibit highly variable kurtosis, confirming the diverse nature of their constituent properties.

In summary, this analysis highlights the profound statistical heterogeneity inherent in gene-level multi-omics data. The prevalence of severe skewness, heavy tails, and considerable variance in these distributional properties underscores the need for a model architecture, such as the one we propose, that is robust to the complex and varied statistical landscapes of multi-omics data.

## B    RATIONALE FOR ASYMMETRIC NODE WEIGHTING STRATEGY

In our model, the BNHC layer applies its non-uniform weighting scheme exclusively to the ontology-derived hypergraph, while using uniform weights for the pathway-derived hypergraph. This appendix details the rationale for this asymmetric design, which is grounded in the fundamental differences in granularity and functional coherence between these two biological data sources.

The primary distinction lies in their construction and scope. Ontology gene sets, such as those from Gene Ontology (GO), are organized hierarchically and often represent broad functional categories that can range from highly specific to very general. A high-level GO term like "signal transduction" have thousands of genes whose functional roles and importance vary significantly. Within such a large, heterogeneous collection, a mechanism is required to identify the core, functionally central genes. In contrast, pathway gene sets from databases like KEGG or Reactome are typically expert-curated to represent specific, well-defined molecular processes. These pathways are generally smaller, and their member genes are considered to be a more functionally homogeneous and tightly coupled module, where each component plays an integral role.

For the broad and heterogeneous ontology gene sets, we use our statistics-based node weighting. This method effectively prioritizes the likely "core" members within a large functional group by identifying those that form a dense local neighborhood in HumanNet v3. For the smaller, more homogeneous pathway gene sets, applying such a statistical test is less informative. We therefore

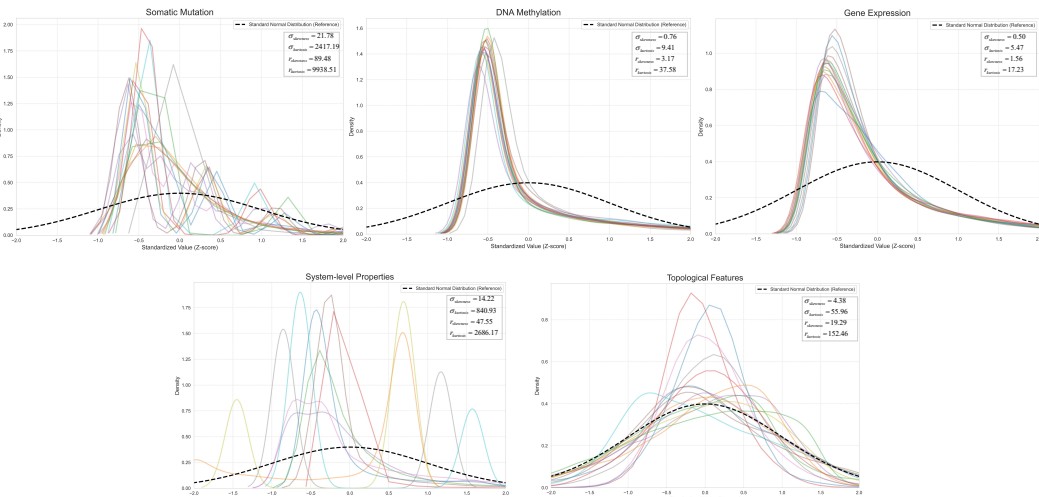

Figure 3: Distributions of multi-omics features. Kernel Density Estimate (KDE) plots for the five feature categories after z-score standardization. The standard normal distribution is shown as a dashed black line for reference. The visualization highlights the significant departures from normality and the profound statistical heterogeneity among different feature types.

use a uniform weight ($w_{ve} = 1$) to respect the expert curation that defines these pathways as cohesive functional units. This tailored, asymmetric approach is a key modeling decision that allows for more nuanced learning from each data source and directly motivates the dual-channel architecture used in our framework.

## C  IMPLEMENTATION DETAILS

All experiments are conducted on an Ubuntu 20.04 server equipped with 256GB of RAM and an NVIDIA A100 80G GPU. The DEMONet framework is implemented in Python 3.6.13, utilizing PyTorch 1.13.0 and PyTorch Geometric 2.3.0 for model construction and training.

**Model Configuration.** In the MSE module, we employ a Random Forest classifier consisting of 200 decision trees, with a maximum depth of 5 for each tree. The resulting one-hot encoded vectors from these trees are concatenated to form the initial sparse feature representation for each gene. For the core network architecture, all BNHC layers in DEMONet utilize a hidden dimension of 256.

**Training Protocol.** The model is trained for 100 epochs using the Adam optimizer. We set the initial learning rate to 2e-4 and apply L2 regularization with a weight decay coefficient of 5e-3. The cross-entropy loss function is used to optimize the model parameters. To ensure reproducibility, our complete source code, datasets, and detailed experimental configurations are provided in the Supplementary Material.

**Evaluation Protocol.** For a fair and robust comparison, we evaluate all models under a stratified five-fold cross-validation scheme. The final performance metrics are reported as the mean and standard deviation across five independent runs, each initiated with a different random seed.

For the baseline methods, we categorize them into two groups. For MTGCN, MODIG, DISHyper, and MNGCL, we utilize their publicly available official implementations and maintain the default hyperparameter settings as recommended by their authors. For 20/20+, EMOGI and DORGE, we re-implemented the models following the descriptions in their original publications. Our implementation of EMOGI achieved slightly better performance than reported in its original paper on our dataset.

# D  ASSESSING DEMONET PREDICTION RESULTS USING LITERATURE-ANNOTATED CANCER GENES

To validate the accuracy of our predictions, we assess the enrichment of literature-annotated cancer genes within the DEMONet ranking results. For this analysis, we utilize a curated gene set from the CancerMine database, a comprehensive resource that extracts cancer gene associations from scientific literature. Specifically, we focus on a high-confidence subset of CancerMine genes, defined as those supported by evidence from five or more publications, to ensure the robustness of our validation.

Our analysis reveals a strong correspondence between DEMONet's ranking results and this literature-based evidence. First, we examine the density of these genes at the top of our ranked list. We observe a remarkable enrichment: among the top 200 genes prioritized by DEMONet, 86 are present in the high-confidence CancerMine set (Figure 4A). This result underscores the high quality and reliability of our model's top predictions.

Second, to analyze the overall distribution, we partition the entire ranked list into ten deciles and count the number of high-confidence CancerMine genes within each. A significant enrichment is observed in the first decile, which contains the highest concentration of these literature-supported genes (Figure 4B). Furthermore, we identify a clear and consistent trend across the deciles: the number of high-confidence cancer genes progressively decreases as the rank decreases. This strong negative correlation between rank and literature-based evidence collectively demonstrates that DEMONet effectively prioritizes genes with established roles in cancer, thereby validating its ability to identify cancer genes.

# E  CHARACTERIZATION OF DEMONET-PREDICTED CANCER GENES

To further investigate the biological mechanisms of our top predictions, we analyze the Top-100 Predicted Cancer Genes (PCGs) from DEMONet for enrichment in three independent functional genomics datasets representing key cancer-driving mechanisms. We compare the enrichment of our PCGs against that of Known Cancer Genes (KCGs), which are expected to be enriched, and Neutral Genes (NGs), which serve as a negative control.

**Association with Transposon-based Gene Inactivation.** First, we assess whether PCGs act as tumor suppressor genes using data from Sleeping Beauty (SB) transposon insertional mutagenesis. This powerful genetic tool identifies potential tumor suppressors by screening for genes whose disruption via transposon insertion drives tumorigenesis. Using a set of candidate genes identified through SB screens (from the SBCDDB database), we perform an enrichment analysis. As expected, KCGs are significantly enriched in these inactivation-driven candidates, whereas NGs show no enrichment (Figure 4B). Crucially, our PCGs also show a highly significant enrichment (P-value = 4.5e-9). This result suggests that DEMONet can effectively identify genes that likely function as tumor suppressors.

**Involvement in Oncogenic Gene Fusions.** Second, we examine the association of PCGs with oncogenic gene fusions, a hallmark of cancer resulting from chromosomal rearrangements. We analyze a comprehensive list of genes involved in fusions across 33 TCGA tumor types (collated from TumorFusions and Gao et al.). The analysis confirms that KCGs are significantly enriched in fusion events compared to NGs. Notably, our PCGs exhibit a similarly strong and significant enrichment (P-value = 2.2e-8, Fisher's exact test), a pattern not observed in randomly selected gene sets (Figure 4C). This finding indicates that DEMONet is capable of identifying driver genes that operate through gene fusion mechanisms.

**Enrichment in Epigenetic Regulators.** Finally, we investigate the link between our predictions and epigenetic regulation, a critical process in tumorigenesis. We test for enrichment against a curated list of 761 Epigenetic Regulator (ER) genes from the EpiFactors database. Our analysis shows that while KCGs are strongly enriched in ERs, NGs are not (Figure 4D). Importantly, our PCGs are also significantly enriched in this set of regulators (P-value = 5.6e-4). This suggests that a subset of the genes prioritized by DEMONet may drive cancer through the dysregulation of epigenetic modifications, highlighting another relevant mechanism captured by our model.

# F    THE USE OF LARGE LANGUAGE MODELS

We utilize a Large Language Model (LLM) as a writing assistance tool in the preparation of this manuscript. The primary role of the LLM is to improve the clarity, conciseness, and grammatical accuracy of the text.

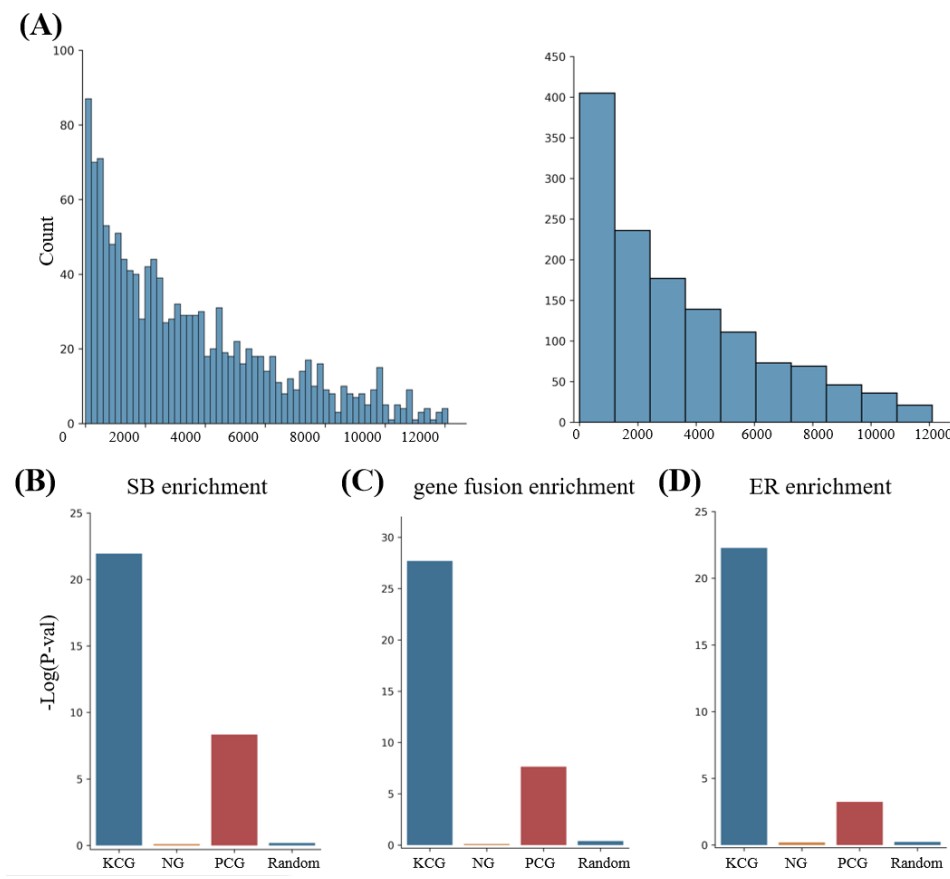

Figure 4: (A) Analysis of DEMONet ranking results based on CancerMine database. On the left, the number of cancer genes enriched in each interval (with an interval size of 200) is depicted, while the right side presents the decile enrichment analysis results. Enrichment analysis of KCGs, NGs, and PCGs in SB inactivating pattern gene list (B), gene fusion list (C), and ER gene list (D).

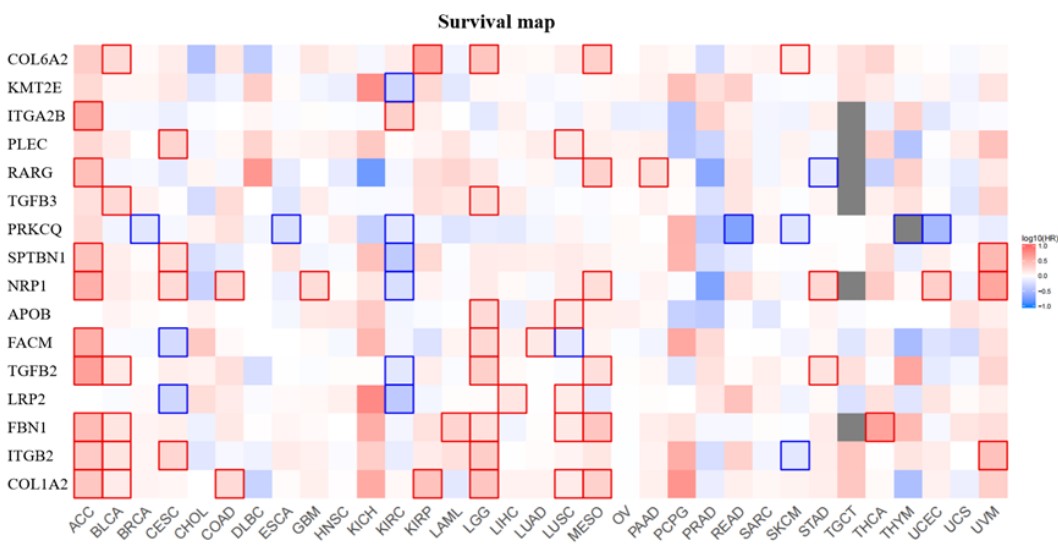

Figure 5: Results of survival analysis for 16 novel cancer genes across 33 cancer types. Points outlined in red or blue highlight genes whose expression significantly impacts the survival duration of specific cancers.

Table 4: The DEMONet-predicted 16 novel cancer genes.

| Rank | Gene Symbol |
|------|-------------|
| 10 | COL6A2 |
| 19 | ITGB3 |
| 22 | FBN1 |
| 29 | LRP2 |
| 32 | TGFB2 |
| 37 | FANCM |
| 38 | APOB |
| 57 | NRP1 |
| 63 | SPTBN1 |
| 65 | PRKCQ |
| 66 | TGFB3 |
| 68 | RARG |
| 77 | PLEC |
| 78 | ITGA2B |
| 88 | KMT2E |
| 98 | COL6A2 |

