# OpenReview forum: "DEMONet: A Dual-channel Multi-omics Integration Hypergraph Network for Cancer Gene Identification"
_ICLR.cc/2026/Conference — ICLR 2026 Conference Withdrawn Submission_

### Official Review · Reviewer_aRmW · 2025-10-30

**Soundness:** 2
**Presentation:** 3
**Contribution:** 2
**Rating:** 2
**Confidence:** 4

**Summary:**

To address the limitations of Hypergraph Neural Networks (HGNNs) in multi-omics feature integration—including inability to handle data heterogeneity, failure to capture the contextual functional specificity of genes due to the model’s assumption of node homogeneity within hyperedges, and poor performance of simple early fusion for multi-omics features—this study proposes DEMONet. This network comprises three core modules: a tree-based sparse encoder, a biologically-informed node-weighted hypergraph convolutional layer, and a dual-channel hypergraph network architecture. The proposed method is partially verified through comparative experiments with multiple baseline methods, ablation studies, and downstream task analyses.

**Strengths:**

1) The writing is clear, the structure is complete, the logic is coherent, and it is easy to follow.
2) The research motivation is clear, and the principle of the proposed method is concise. The paper provides public code and detailed data-processing descriptions.
3) It includes downstream task analysis, enabling the prediction of novel cancer-related genes and thus possessing practical application value. The accompanying functional and survival analyses give preliminary evidence that some of the top-ranked genes are linked to cancer mechanisms and patient prognosis.

**Weaknesses:**

$\textbf{Major}$:

1. $\textbf{Insufficient innovation and verification}$: The proposed method has limited innovation and lacks adequate verification. The core idea of HGNNs has already been applied in DISHyper, and dual-channel designs are also common (e.g., the baseline method MNGCL uses three channels for feature fusion). Additionally, there is content overlap between the Introduction and Related Work sections, and parts of the Introduction bear high similarity to DISHyper.
2. $\textbf{Lack of statistical completeness in experimental results}$: Although statistical significance is indicated when comparing with SOTA methods, no statistical significance is reported for ablation study results. Furthermore, standard deviations are omitted from all experimental results, making it impossible to assess result stability. Additionally, experimental evaluation is somewhat incomplete because it reports only AUROC and AUPRC, omitting ranking-oriented metrics such as precision@K and recall@K, which makes it difficult to gauge the model’s practical prioritization performance.

3. $\textbf{Inadequate verification of biologically-informed node weights}$: The authors claim that biologically-informed node-weighting can reflect gene heterogeneity, but no supporting ablation experiments are provided. It is recommended to supplement comparative experimental results between this weighting strategy and unweighted graphs (without node weights) to verify the effectiveness of the weights.
4. $\textbf{Unproven balance between model complexity and benefits}$: As shown in Table 3, the dual-channel variant improves AUROC and AUPRC by less than 0.013 compared with using either a single ontology graph or a concatenated hypergraph. Such a small margin might stem from the additional parameters rather than from a clear structural advantage, leaving some uncertainty about the necessity of maintaining two separate channels. The dual-channel architecture may potentially increase model complexity compared to single-channel methods. However, the authors do not provide quantitative metrics such as running time or peak memory usage. This makes it impossible to determine whether the likely substantial increase in complexity is justified, especially considering that the improvement in model performance appears to be only marginal.
5. $\textbf{A pronounced generalization gap is evident}$: The cross-validated AUPRC is roughly 0.90, whereas it falls below 0.15 on the external OncoKB and below 0.1 on the ONGene sets, indicating possible over-fitting to the training distribution and limited utility for novel driver-gene prioritization.
6. $\textbf{Unknown impact of Random Forest encoding parameters}$: The impact of the Random Forest parameters used for constructing sparse features on encoding results and subsequent model performance is not analyzed.
7. DEMONet adopts a two-stage pipeline of "Random Forest encoding --> HGNN feature extraction," where encoding performance directly affects subsequent feature extraction. It is recommended to explain why neural network encoding (to enable end-to-end training) is not used and compare the performance differences between the two-stage and end-to-end designs.
8. Does encoding with random forests lead to the loss of information in the original data?

$\textbf{Typos}$

Variables "g, v, e" should be formatted as mathematical symbols "$g$, $v$, $e$" instead of plain text.

**Questions:**

Please see the weaknesses.

---

### Official Review · Reviewer_Wq7y · 2025-11-01

**Soundness:** 3
**Presentation:** 3
**Contribution:** 2
**Rating:** 6
**Confidence:** 4

**Summary:**

The authors propose DEMONet, a framework designed to address heterogeneity in multi-omics data by extracting robust and sparse features using a random forest. They further introduce a Biologically-Informed Node-Weighted Hypergraph Convolutional Layer (BNHC) to incorporate biological context into hypergraph representations. Specifically, BNHC assigns biologically informed prior weights to nodes based on their local connectivity with other members of a hyperedge in HumanNet_v3, rather than using uniform weights during node-weighted aggregation. In addition, DEMONet applies hyperedge-level weighting during node aggregation according to the proportion of known cancer driver genes within each hyperedge. The authors validate the effectiveness of their approach using two independent test datasets.

**Strengths:**

- The authors propose a simple yet effective method for cancer gene identification, and each proposed module is well-founded with clearly articulated motivations.

- They also perform comprehensive ablation studies to validate the contribution of each module.

**Weaknesses:**

- The data source section lacks sufficient detail. The authors should specify the dimensionality of each multi-omics feature and clearly describe the biological meaning or type of information each feature represents.

- While the authors conduct ablation studies on features derived from MSE, this is not the only available feature extraction approach. They should include comparisons with standard methods such as PCA to provide a fair baseline.

- A sensitivity analysis on key hyperparameters, such as the number of layers, should be included to evaluate the robustness of the model.

- The authors should also provide a running-time analysis to demonstrate the computational efficiency of the proposed method.

**Questions:**

Please address the following weaknesses:

- Could you specify the dimensionality of each multi-omics feature and explain the biological meaning of each feature type?

- The ablation studies are conducted only for features derived from MSE. Could you include comparisons with other basic feature extraction methods such as PCA to strengthen your evaluation?

- Could you provide sensitivity analysis results for key hyperparameters (e.g., the number of layers) to assess model robustness?

- Please include a running-time analysis to demonstrate the computational efficiency of your proposed method.

---

### Official Review · Reviewer_VyN8 · 2025-11-01

**Soundness:** 2
**Presentation:** 2
**Contribution:** 2
**Rating:** 2
**Confidence:** 4

**Summary:**

A hypergraph neural network called DEMOnet was developed to integrate multiple modalities for the cancer gene identification task. This model is a biologically informed hypergraph neural network that weights the importance of nodes based on biologically (and statistically) meaningful criteria.

**Strengths:**

* The paper is well written, easy to follow.
* Authors provided various experiments
* Authors integrated biologically meaningful information in a valid manner.

**Weaknesses:**

* It appears that the experiments were conducted only once and the performance was reported based on that single run. To ensure that the model’s performance is not dependent on a specific random seed but is statistically meaningful, it is necessary to repeat the experiments multiple times and report the mean and standard deviation of the performance.
* It is necessary to evaluate the performance of a traditional attention-based hypergraph neural network without biologically informed weighting. If the model performs well even without the biologically informed weighting and relies solely on attention, it may suggest that the biologically informed approach is not strictly necessary.
* It is possible to apply traditional hypergraph neural networks to this dataset. Therefore, instead of comparing only with baselines specific to this field, basic hypergraph neural networks should also be included as baselines when reporting performance. The authors should additionally report the performance of models such as HGNN[1], AllSetTransformer[2], SHINE[3] and Natural-HNN[4,5].

[1] Hypergraph Neural Networks, AAAI 19

[2] You are AllSet: A Multiset Function Framework for Hypergraph Neural Networks, ICLR 22

[3] SHINE: SubHypergraph Inductive Neural nEtwork, NeurIPS 22

[4] Capturing functional context of genetic pathways through hyperedge disentanglement, ICLR 25 workshop MLGenX

[5] Disentangling Hyperedges through the Lens of Category Theory, NeurIPS 25

**Questions:**

Identical to the 'Weakness' part.

---

### Official Review · Reviewer_WbAa · 2025-11-01

**Soundness:** 2
**Presentation:** 3
**Contribution:** 1
**Rating:** 2
**Confidence:** 4

**Summary:**

The paper “DEMONet: A Dual-channel Multi-omics integration hypergraph network for cancer gene identification” used a multi-omics integration hypergraph neural network to identify cancer genes. The study proposed three methods: (1) a multi-omics encoding module, which is a tree-based sparse encoder, aims to increase the dimension by a Random Forest to decrease the complexity of each feature; (2) a biologically informed node-weighted hypergraph convolutional (BNHC) layer; (3) a dual-channel hypergraph network architecture. The results outperformed the existing state-of-the-art methods. The survival analysis also revealed that the identified genes are strongly correlated with certain cancers.

While the biologically informed node-weight calculation represents a notable contribution to cancer gene identification, the study does not include an ablation analysis to evaluate the specific impact of this weighting scheme. Moreover, the hypergraph architecture has already been explored in DISHyper (Deng et al., 2024). Overall, the work introduces several components, but its methodological novelty is largely incremental relative to DISHyper, with most architectural ideas extending or combining existing approaches rather than fundamentally new model designs.

**Strengths:**

[S1] Thoughtfully integrated methodological elements: The biologically-informed node-weighting strategy is a meaningful design choice that reflects the varying importance of genes across cancers. While conceptually straightforward, this weighting mechanism helps ground the model in biological relevance and provides a clear motivation for focusing on connected genes within local structures. The hyperedge-to-node aggregation also leverages this weighting to emphasize differential contributions, which is a reasonable and interpretable design for this problem domain.

[S2] Solid results: The results are clear, supported by clinical relevance and an ablation study that demonstrates the contribution of each component. Additionally, the model achieves SOTA results in cancer-driver score prediction.
.

**Weaknesses:**

[W1] Incremental novelty in methodology: Although well-executed, the core architectural components, including dual-channel processing and the hypergraph architecture, largely extend established frameworks rather than introducing fundamentally new modeling concepts. For example, the aggregation mechanism closely follows prior work (e.g., Deng et al., 2024), and the dual-channel setup resembles common multi-view fusion strategies.

[W2] Contribution framing and scope: The paper would benefit from framing its contribution as a biologically motivated integration rather than a new architecture. The strength lies in combining existing ideas with strong biological validation, but the work offers limited methodological advances beyond this specific application domain.

[W3] Ablation study and weight distribution analysis: There is no ablation study regarding the significance of the weighting algorithm. Additionally, more properties of the weights, such as their distribution and value range, should be included to demonstrate their effectiveness.

[W4] Limited interpretability:
While DEMONet performs well, the paper provides little insight into why it works—e.g., which features or node weights drive predictions. Including interpretability or case analyses would make the findings more meaningful for biological understanding.

**Questions:**

[Q1] Clarification of differences from prior works: Please clarify how the aggregation mechanisms differ from those in the existing message-passing formulations of conventional HGNN (Feng et al., 2019) and prior work (Deng et al., 2024).

[Q2] Ablation study and statistical analysis of node weights: What is the result of ablation if all node weights are set to 1? Please perform an ablation study of biologically informed node weights to quantify their contribution to model performance. It would be helpful if the authors could report basic statistics (e.g., range, mean, variance, or histogram) of the learned or pre-computed node weights to show how they behave across different genes or samples, and include this in the Supplementary Materials.

[Q3] Clarification of color encoding in Appendix: What is the definition of the gray color in Appendix Figure 5?

[Q4] Code and reproducibility: Please include a complete README and code for the node-weight calculation to ensure full reproducibility of the results and weighting process.

---

### Note · Authors · 2025-11-12

I have read and agree with the venue's withdrawal policy on behalf of myself and my co-authors.